# Effects of Pilates Matwork Core Exercises on Functioning in Middle-Aged Adult Women with Chronic Nonspecific Low Back Pain Through Flexion Relaxation Phenomenon Analysis: A Pilot RCT

**DOI:** 10.3390/jfmk10040433

**Published:** 2025-11-06

**Authors:** Nicola Marotta, Alessandro de Sire, Federica Pisani, Michele Mercurio, Ennio Lopresti, Lorenzo Scozzafava, Andrea Parente, Giorgio Gasparini, Umile Giuseppe Longo, Antonio Ammendolia

**Affiliations:** 1Physical Medicine and Rehabilitation, Department of Experimental and Clinical Medicine, University of Catanzaro “Magna Graecia”, 88100 Catanzaro, Italy; nicola.marotta@unicz.it; 2Research Center on Musculoskeletal Health, MusculoSkeletalHealth@UMG, University of Catanzaro “Magna Graecia”, 88100 Catanzaro, Italy; alessandro.desire@unicz.it (A.d.S.); gasparini@unicz.it (G.G.); ammendolia@unicz.it (A.A.); 3Physical and Rehabilitative Medicine, Department of Medical and Surgical Sciences, University of Catanzaro “Magna Graecia”, 88100 Catanzaro, Italy; federika.pisani@libero.it (F.P.); ennio.lopresti@studenti.unicz.it (E.L.); lorenzo.scozzafava@studenti.unicz.it (L.S.); parenteandrea3@gmail.com (A.P.); 4Department of Orthopaedic and Trauma Surgery, “Renato Dulbecco” University Hospital, “Magna Græcia” University, 88100 Catanzaro, Italy; 5Fondazione Policlinico Universitario Campus Bio-Medico, 00128 Roma, Italy; g.longo@policlinicocampus.it; 6Research Unit of Orthopaedic and Trauma Surgery, Department of Medicine and Surgery, Università Campus Bio-Medico di Roma, Via Alvaro del Portillo 21, 00128 Roma, Italy

**Keywords:** pilates, chronic non-specific low back pain, flexion relaxation phenomenon analysis

## Abstract

**Objectives**: Pilates is frequently recommended for patients with Chronic Nonspecific Low Back Pain (CNLBP) due to its potential to enhance posture, muscle strength, trunk flexibility, and stability. However, to date, there is no robust evidence supporting the effectiveness of Pilates in managing CNLBP. This study aimed to assess the effects of 8 × 8 Pilates Matwork core exercises on pain and functioning in middle-aged adult women with CNLBP, through a flexion relaxation phenomenon (FRP) analysis. **Methods**: We included middle adult women (n = 21) with diagnosis of CNLBP and a Numeric Rating Scale (NRS) > 4. The experimental group underwent a treatment of eight Pilates Matwork sessions, biweekly for 4 weeks, lasting about 40 min. The control group underwent standardized exercises used for managing CNLBP. Outcome measures included NRS, Oswestry Disability Index (ODI), Quebec Back Pain Disability Scale (QBPDS), and the FRP ratio via surface electromyography during trunk maximum flexion. We evaluated the participants at the baseline (T0), at the end of the 4-week treatment (T1), and at 4 weeks after the end of the treatment, at 8 weeks from the baseline (T2). **Results**: In this pilot RCT, 21 middle-aged adult women affected by CNLBP were randomly allocated with a ratio of 1:1 in the Pilates group, and in parallel in the control group. The experimental group showed a significant improvement in ODI and QBPDS scores compared to the control group, maintained at follow-up for ODI, along with an NRS reduction at T2. About FRP, Pilates has proven to be comparable to conventional treatment, showing no significant difference in FRR at T1 and T2. Only the experimental group exceeded the 9.5 cutoff at T2, as a protective predictive index for CNLBP. **Conclusions**: This pilot RCT provided preliminary evidence that Pilates might be an effective rehabilitation method, enhancing functioning and pain management in middle-aged adult women affected by CNLBP. The FRP study proves to be efficient in translating clinical assessments into rehabilitation assessment measures.

## 1. Introduction

Low back pain is characterized by localized pain in the anatomical region between the margin of the 12th rib and the iliac crests; if the pain persists for more than 12 weeks, it is considered chronic [1,2]. It has been estimated that the one-time prevalence of low back pain (LBP) could be as high as 84%, while the prevalence of Chronic Nonspecific Low Back Pain (CNLBP) is about 25% [3].

CNLBP is the most common and widely studied musculoskeletal problem worldwide; thus, in industrialized societies, it is the leading cause of disability and work absenteeism among people younger than 45 years of age [4]. Moreover, the incidence of this condition increases significantly with age, and several lifestyle factors such as obesity, physical inactivity, and smoking. To date, LBP possesses a significant socioeconomic burden as well as a challenge for the health care system [5].

Beyond supportive pharmacological treatments, the most employed strategy to manage this condition is exercise, which is the most widely prescribed intervention worldwide and recognized as beneficial for people with CNLBP [6]. The quality of evidence on clinical effectiveness ranged from low to moderate in demonstrating the improvements in pain intensity and disability after various interventions, which also included the following: physiotherapy, manual therapy, spinal orthoses, pharmacological and nutraceutical interventions, and interventional treatments, including oxygen–ozone therapy and radiofrequency [7,8,9,10,11,12,13]. In this scenario, a systematic review with meta-analysis showed that all exercises except stretching for pain and the McKenzie method for disability were effective in reducing pain and disability in patients with CNLBP. Pilates led the most effective intervention for reducing both pain and disability, followed by mind–body exercises for pain and strength and core-based exercises for disability [14].

In the context of low back pain, the flexion relaxation phenomenon (FRP) is a predictive indicator, as it consists of the presence of myoelectric silence in full trunk flexion; FRP can be quantified using surface electromyography on the paraspinal muscles [15,16,17].

It also seems to be influenced by yoga asanas postures in women with CNLBP, where results indicated positive effects on pain intensity and helped improve the management of the condition [18].

Cuddy et al. showed Pilates might be tailored to individual patient needs, reducing kinesiophobia, and enhancing body awareness with muscle strengthening core exercises [19]. Moreover, Tottoli et al. demonstrated the effectiveness of Pilates when compared to home exercise for pain, disability, and function in individuals with CNLBP [20].

Pilates might emphasize the activation of the trunk and lower back-stabilizing muscles; so, over the last decade, Pilates has often been prescribed for patients suffering from LBP. Nevertheless, up to date, there is no robust evidence supporting the benefit of Pilates in the management of CNLBP. Even though Pilates is widely used to improve posture, muscle strength, flexibility, and stability, very few studies have rigorously evaluated its effects using instrumentation such as sEMG and inertial motion units.

Therefore, by the present pilot randomized controlled trial, we aimed to fill this gap in the scientific literature by assessing the effects of 8 × 8 Pilates Matwork core exercises on pain and functioning in adult women with CNLBP through an FRP analysis; as a second hypothesis, it also served us indirectly to compare Pilates to conventional methods.

## 2. Materials and Methods

### 2.1. Participants

This pilot RCT included adult women, recruited at the Physical Medicine and Rehabilitation Unit of the AOU “Renato Dulbecco” of Catanzaro, which followed the subsequent inclusion criteria: (i) middle-aged adult women with age ranging from 40 to 65 years; (ii) diagnosis of CNLBP; (iii) Numeric Rating Scale (NRS) > 4; (iv) consent to discontinue pharmacological treatment with NSAIDs, muscle relaxants and any other therapy that may interfere with the study assessments; and (v) willingness to not reintroduce such therapies for the entire duration of the study.

Moreover, we followed the exclusion criteria: severe cognitive decline (Mini-Mental State Examination score < 24) and/or patients unable to provide informed consent; acute back pain that has recently worsened clinical presentation; pregnancy or breastfeeding; current tumor pathologies; concomitant performance of other rehabilitation and/or physical instrumental therapies; previous surgical procedures within six months; and known allergy to the adhesives used to position the markers.

The study was approved by the Regional Ethics Committee Regione Calabria number 64/2024), and was conducted according to the Declaration of Helsinki.

### 2.2. Intervention

The patients were randomly assigned to two groups: the experimental group and the control group. Participants were randomly allocated to the two study groups (intervention or control), using a method of simple randomization. The assignment sequence was generated by a statistical software package (The jamovi project (2025). jamovi (Version 2.6) [Computer Software]. Retrieved from https://www.jamovi.org), ensuring that each participant had an equal and independent probability of being assigned to either study arm. The generated sequence was not restricted or manipulated for balancing purposes.

Patients in the control group underwent a standardized and supervised set of exercises commonly used by physical therapists for managing chronic low back pain [21], for an eight-session program every two weeks, for four weeks. The exercises included the following: (1) indoor-bike for 10 min, (2) lower limb stretches, (3) lumbar twists (lie supine on mat and twist both knees side to side), (4) bridges (lie supine on mat and lift the buttocks off the mat), (5) lumbar flexion (lie supine and roll the knees to the chest), (6) cobra (lie prone and do a half push-up from the waist), (7) ball crunches (lift the pelvis up as before, then roll the ball towards the buttocks, roll way again and lower), (8) trunks twists and trunk side-bends (standing with feet shoulder-width apart, twist trunk around side to side and let arms swing, then side-bend the trunk alternating left and right) [21].

The patients in the experimental group underwent a treatment that consisted of eight Pilates Matwork sessions in total, biweekly for a period of 4 weeks, lasting about 40 min, carried out without the use of equipment.

We evaluated the participants at the baseline (T0), at the end of the 4-week treatment (T1), and at 4 weeks after the end of the treatment, at 8 weeks from the baseline (T2).

### 2.3. The Experimental 8-Pilates Matwork Exercise Program [22,23,24]

“Spine stretch”: The subjects began in a seated position at a 90-degree angle, with legs extended forward, shoulder-width apart, feet in a hammer position, arms extended parallel to the legs and the ground. The participants inhaled to prepare for the movement. During exhalation, they bent their torso forward, trying to bring their navel towards the spine through abdominal contraction, lowering their head towards their knees with arms extended in front of the body trying to form a C with the spine. Once at the point of maximum flexion, we recommended inhaling to prepare for the return. Exhaling, the subject returned to the starting position, realigning the spine with the hips.“Spine twist”: The participants started from a seated position at a 90-degree angle, with legs spread out extended forward, feet in a hammer position, arms abducted at 90°, externally rotated and shoulders away from the ear. Before starting with the trunk rotation, it was important to lengthen the spine upwards even more. The participant began with the trunk rotation exhaling, bringing one upper limb backward, with the head following the movement of the hand stretching back, without moving the pelvis or body weight and performing a dorsal twist, ten times per side.“Hundred”: Starting from a supine position, the subject raised his legs and bent his knees at 90°, keeping the abdomen contracted and the navel towards the spine. The spine was necessary to adhere to the ground as much as possible. The participant bent his torso forward until the lower angle of the scapula touched the ground, then the arms were detached from the ground, keeping them with the palm facing down for two inhalations, then, the palm of the hand was rotated upwards to perform two exhalations. Swing the arms energetically and in a coordinated way with the breathing rhythm.“Leg changes”: the subject started from a supine position, keeping the abdomen and pelvis in a neutral position, with arms along the trunk and palms of the hands on the floor. The hips were bent, bringing the thighs towards the abdomen. The participant had to inhale to prepare for the movement, further stabilizing the pelvis. Subsequently, it was necessary for the subject to exhale bringing one leg towards the abdomen, trying to adhere to the ground with the lumbar area, then, instead of inhaling, going to touch the ground with the foot and return to the starting position. It was essential to alternate the movement with both legs, performing ten repetitions per leg. The same exercise could be performed keeping both feet raised, in the same position as the Hundred exercise, alternating the movement of the legs bringing the foot to the ground.“Roll up”: The subject began lying on their back with legs slightly spread and arms extended behind their head. They inhaled to prepare for the movement. They exhaled, lifting their arms, head, and back, slowly lifting off the mat one vertebra at a time, “unrolling” the spine, to engage the core and pull the navel in. Once in a seated position at 90°, the participant was instructed to inhale and, exhaling, bend the torso over the thighs (as in the Spine stretch exercise). At the point of maximum flexion, they also inhaled and, exhaling, returned to the starting position, slowly resting on the ground one vertebra at a time.“Pelvic curl”: Starting position lying on their back with knees bent and feet flat on the ground. The participant aligned their nose with their navel and stabilized the pelvis. They kept the spine in a neutral position, inhaled to prepare for the movement, and exhaled while contracting the abdomen, bringing the navel towards the spine and trying to flatten the lumbar curve, making the spine completely adhere to the mat. They were recommended to inhale while maintaining the position, then exhale and relax the muscles, returning the spine to neutral, recreating the lumbar lordosis.“Shoulder bridge”: lying on their back with knees bent and feet flat on the ground, to align the nose with the navel and stabilize the pelvis. It was necessary to inhale to prepare for the movement; exhaling, they lifted the pelvis and then the spine from the mat, trying to “unroll” the spine from the ground, vertebra by vertebra. Once in the final position, the patient had to inhale contracting the glutes. Exhaling, return to the starting position, making the spine adhere to the ground one vertebra at a time.“Leg circles”: The patients started lying on their back, bending the hip at 90°. The knee of the flexed hip remained completely extended, then, the participant was instructed to draw a circle with the extended leg, starting the movement from the hip, activating the core and trying to keep the rest of the body still. For each gesture performed, they inhaled, and exhaled with the next. Ten repetitions clockwise and ten counterclockwise. Then, repeat the exercise with the other leg.

The eight selected exercises were performed for ten repetitions, preceded by a training phase for the correct execution of the same exercise and breathing for the entire duration of the session, under the guidance of an expert physiotherapist (Figure 1).

### 2.4. Outcome Measures

-The Oswestry Disability Index (ODI) is a self-administered questionnaire used to measure pain and disability in patients with chronic low back pain (LBP). It takes a few minutes to complete and is divided into ten sections that assess how LBP affects different aspects of daily life, such as pain intensity, personal hygiene, lifting weights, walking, sitting, standing, sleeping, sex life, social life, and traveling. Each section has six response options, with scores ranging from 0 to 5, where 0 represents no difficulty or pain and 5 represents inability to perform the activity or disabling pain. The ODI produces a final functional score ranging from 0 to 100, interpreted as follows: 0–20% minimal disability without therapy needed; 20–40% moderate disability requiring conservative therapy; 40–60% severe disability requiring further examination; 60–80% devastating disability requiring substantial intervention and bedridden.-The Quebec Back Pain Disability Scale (QBPDS) is a self-assessment tool developed in 1995 to assess disability related to low back pain. It consists of 20 items that measure an individual’s ability to perform various daily activities, including walking, sitting, lifting objects, and bending. Each item has six possible responses, with scores from 0 to 5, where 0 indicates no difficulty and 5 indicates severe disability. The total score ranges from 0 (no difficulty) to 100 (maximum disability). The score is classified as follows: 0/20% no or mild disability; 21/40% moderate disability; 41/60% severe disability; 61/80% very severe disability; 81/100% extremely severe disability.-NRS: This is a unidimensional, quantitative, 11-point numerical rating scale for assessing and quantifying pain. The scale requires the operator to ask the patient to select the number that best describes the intensity of their pain, from 0 to 10, at that precise moment. The patient is asked: “If 0 represents no pain and 10 represents the worst possible pain, what is the pain you are experiencing now?”-Relaxation/Flexion Phenomenon assessment: The forward bending/extension movement of the torso was evaluated while the patient stood upright. The patient was instructed to move in response to a verbal command, keeping their knees straight but not locked and their arms hanging freely. They were asked to slowly bend forward to full flexion over 3 s, pause for 3 s at full flexion, and then return to the starting upright position over 3 s. This movement was performed three times, and the average of the tests was used in the analysis. Before the first test, patients practiced the assessment to become familiar with the movement. Surface electromyography was used to assess muscle activation. A four-channel conditioning module (BTS FREEEMG 1000) with a common mode rejection ratio greater than 100 dB and a 20–450 Hz band-pass filter amplified the signal 2000 times using a sampling frequency of 1 kHz and wireless transmission. The signals were captured using self-adhesive, single-use Ag/AgCl surface electrodes, 1 cm in diameter. After cleaning the skin with 70% alcohol, the electrodes were placed 2 cm apart, center to center, on the paravertebral muscles at the level of L1–L2 and on the multifidus muscles at the level of the L4-L5 vertebrae on each side. The electrodes were placed with a vertical distance of about 1 cm between their edges while the trunk was in a semi-flexed position. The EMG signal was collected during this motion, and a flexion–relaxation ratio (FRR) was calculated using the approach of Ritvanen et al. [25]. The FRR was calculated as the ratio between the RMS activity during trunk flexion and the RMS activity during full flexion. In this scenario, EMG activity was high during full flexion, which is typical among patients with LBP; conversely, the presence of a flexion–relaxation phenomenon indicates myoelectric silence in full flexion.

### 2.5. Statistical Analysis

To assess the normality of the data, the Shapiro–Wilk test was used. The mean differences between groups were determined using the independent t-test for normally distributed data. Within-group differences were assessed using the paired-samples t-test. Differences across multiple time points within the same group were analyzed using a two-way mixed Repeated Measures Analysis of Variance (RM-ANOVA) to assess changes in the dependent variable over time. To report the magnitude of effects from the mixed model, the Generalized Eta Squared (η^2^_gen_) was utilized. Post hoc comparisons were performed using Holm correction. The raw electromyography signal underwent processing, beginning with the application of a band-pass filter set from 10 Hz to 500 Hz to effectively remove motion artifacts and extraneous low- and high-frequency components. Subsequently, the signal was subjected to full-wave rectification. The Root Mean Square (RMS) value was then calculated from this processed signal using a sliding window approach, employing a window size of 100 milliseconds (ms) and a 50-milliseconds (ms) overlap (equivalent to a 50% overlap). Flexion Relaxation Ratio is the ratio between the energy value delivered by the lumbar muscles during the forward flexion (FLEX) movement of the trunk and the residual energy of the muscles in the phase of maintaining maximum voluntary flexion (MVF). This can be calculated using the following formula: FRR = FLEX/MVF.

In this context, the presence of a flexion relaxation phenomenon indicates myoelectric silence. Consequently, a lower MVF value corresponds to a higher FRR. Neblett et al. stated that the flexion–relaxation ratio (FRR  =  FLEX/MVF) correctly classified 83.5% of patients and 96.7% of control subjects [26]. An FRR  ≥  9.5 could distinguish between chronic NSLBP and control subjects; as the FLEX/MVF ratio increases, myoelectric silence increases, hence the consequent presence of the FRP [18]. IBM SPSS Statistics software (version 26, IBM Corp., Armonk, NY, USA) and G*Power (version 3.1.9.2, Institut für Experimentelle Psychologie, Heinrich Heine Universität, Düsseldorf, Germany) were used for database construction and statistical analysis. A *p*-value of less than 0.05 was considered significant.

## 3. Results

We included a group of female volunteers consisting of 21 women affected by CNLBP allocated with a ratio of 1:1 in a group of Pilates protocol (mean age 53.8 ± 5.5 years; BMI 23.95 ± 1.9 Kg/m^2^), and in parallel in a control group of conventional exercises (mean age 50.8 ± 5.3 years; BMI 24.3 ± 1.7 Kg/m^2^), as depicted in Figure 2.

We assessed baseline ratings of FRR, ODI, QBPDS, and NRS between groups, at each time point, and between individual repeated measures; time points are represented in Table 1.

At the end of the treatment, we demonstrated a significance between group difference for the functioning indices, namely ODI and QBPDS, respectively, with an MD of −11.37 (ES: 0.88) and −7.25 (ES: 0.73). Furthermore, the difference was confirmed at follow-up for the ODI with a maintained MD of −11.75, despite a lower effect size of 0.59. Lastly, we showed a significant difference for perceived pain at T2, with a statistically significant difference in MD = −1.16 (Figure 3).

For all outcomes examined, we reported an improvement trend with significant differences for ODI, QBPDS, and NRS. For the FRR, there was no significant difference at T1; at T2, however, only the experimental group exceeded the cutoff of 9.5 of Neblett et al. [26], as a protective predictive ratio.

## 4. Discussion

The present pilot study aimed to investigate the effects of Pilates Matwork exercises on the FRP and pain intensity in adult women with CNLBP. Our findings revealed significant improvements in terms of functioning, with a significant reduction in disability and pain also suggesting a benefit in the neuromuscular response during trunk flexion.

We observed a significant difference between the experimental and control groups in ODI and QBPDS scores at the end of the intervention. This difference was sustained at follow-up for the ODI; additionally, we identified a significant reduction in perceived pain at T2. Hence, there was an overall trend of improvement across all assessed outcomes in the experimental group compared to control group, particularly in ODI, QBPDS, and NRS. Pilates has proven to be comparable to conventional treatment, since there was no significant difference in FRR at T1 and T2; nevertheless, only the experimental group surpassed the cutoff of 9.5 proposed by Neblett et al. [26] as a protective predictive measure. In this light, Pilates is confirmed as a rehabilitative intervention comparable to a conventional approach, demonstrating a greater protective index at follow-up and significant improvement on functioning and pain indices, even if the small sample size represents a limit on the static power and generalizability of the data obtained [14].

One previous survey of opinions from 30 expert physical therapists underlined the interrelated benefits of Pilates as a form of exercise highly adaptable to patients’ needs; this rehabilitative approach might increase engagement, body awareness, and core muscle strength [19].

Tottoli et al. [20] conducted a study to compare the effectiveness of Pilates with home exercise on pain, disability, and function in patients with CNLBP, reporting a significant reduction in pain intensity and disability in patients undergoing rehabilitation training through Pilates sessions (two per week, 50′ in duration, for six weeks) compared to patients in the control group (undergoing the performance of traditional exercises at home).

Nonetheless, there are still some remaining gaps on neuromuscular activity and response during trunk flexion [27]. In this scenario, the myoelectric silence of the spinal erector muscles during the maintenance of full trunk flexion seems to be an indicator of a proper load shared by the spine [28]. This might indicate that the nature of the support from the spinal erector muscles for the spine when entering a flexion position is different, with significant implications for the mechanical loading and stability of the spine itself [29].

FRP represents a neuromuscular response which was documented during a complete flexion of the trunk in the case of the absence of myoelectric activities in the lumbar muscles. It generally denotes a process of muscle relaxation with subsequent redistribution of the load in the lumbar spine, reducing the active involvement of the lumbar muscles while other structures of the spine, intervertebral discs, and ligamentous structures may take a greater shear stress [30]. This mechanism could explain how trunk flexion affects the total load on the spine and how, possibly, Pilates training can modify the neuromuscular response in a way that promotes more efficient load management and pain reduction in CNLBP [31]. Our findings might, therefore, reflect new evidence regarding the neurobiological mechanisms that could explain the benefits of the rehabilitative use of Pilates matwork exercises in patients with CNLBP.

The FRP, expressed as changes in the flexion–relaxation ratio, was not significantly different in the Pilates group compared to the control group; on the other hand, we observed an increase in FRR after the intervention above the cutoff established by Neblett et al. [26], indicating that Pilates training induced a more efficient neuromuscular response during trunk flexion movements. This could encourage benefits in individuals with CNLBP, including sensitive spinal stability and a more optimal management of muscle loads [32]. Indeed, better neuromuscular coordination and more controlled flexion might decrease the load on the muscles of the lumbar spine, with a consequent reduction in fatigue and overload risk [33].

Pilates appeared to relieve mechanical stress on the spine, by emphasizing core stabilization, improving trunk mobility, and neuromuscular balance [34]. In this context, being able to act on decreasing the intensity of pain not only consequently improves the performance of motor activities but can also facilitate the physical activity into the activities of daily living.

Despite the positive result of our pilot RCT, this study is not free from limitations. Firstly, a relatively small sample size impacts the generalizability of the results. Secondly, our study was characterized by a short follow-up to assess long-term effects. Thirdly, a rigorous non-inferiority trial has not yet been performed to definitively establish the comparative effectiveness. Fourthly, the analysis of pain relief may not be satisfied by a muscle balance and index such as FRR; on the other hand, Neblett et al. [26] established that FRR can provide a useful and sensitive tool to distinguish between patients with NSLBP and control subjects. Lastly, the lack of blinding might be a potential source of bias. Another limitation was the lack of assessment of the patients’ mental status, a factor that could have influenced the treatment decisions [35,36]. In addition, it has already been associated with impaired functional recovery after various procedures [37].

## 5. Conclusions

Pilates is an approach widely used to improve posture, muscle strength, flexibility, and stability; despite low evidence, we have rigorously evaluated its effects through sEMG and inertial motion units. The current study showed that Pilates matwork core exercises could have a significant effect in terms of functioning and pain relief compared to a standard physiotherapy program in a cohort of middle-aged adult women with CNLBP. Moreover, the FRP could be considered as a good predictive indicator of the activation of the paraspinal muscles, albeit showing no differences between Pilates and standard physiotherapy groups. Further high-quality studies with larger samples and longer follow-ups are needed to improve the evidence supporting the efficacy of Pilates in the management of CNLBP. Crucially, only the Pilates intervention group surpassed the threshold of 9.5 for the FRR) at follow-up. Nevertheless, future studies must incorporate larger sample sizes and extended follow-up periods to enhance the evidence base supporting the efficacy of Pilates in the management of Chronic Nonspecific Low Back Pain.

## Figures and Tables

**Figure 1 jfmk-10-00433-f001:**
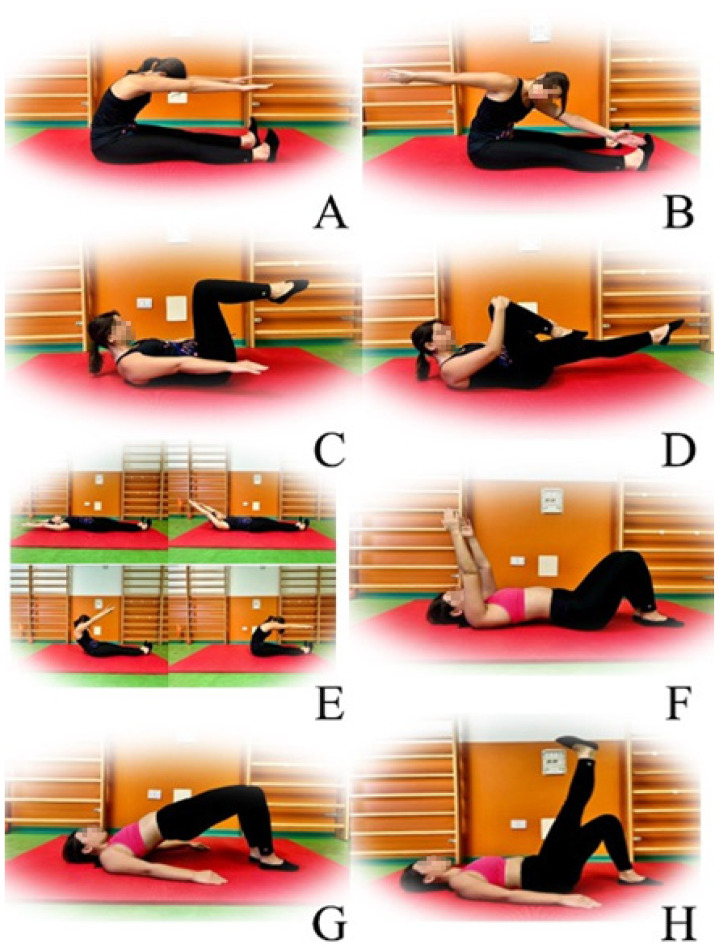
Eight Pilates matwork exercises. (**A**): Spine stretch, (**B**): Spine twist, (**C**): Hundred, (**D**): Leg changes, (**E**): Roll-up, (**F**): Pelvic curl, (**G**): Shoulder bridge, (**H**): Leg circles.

**Figure 2 jfmk-10-00433-f002:**
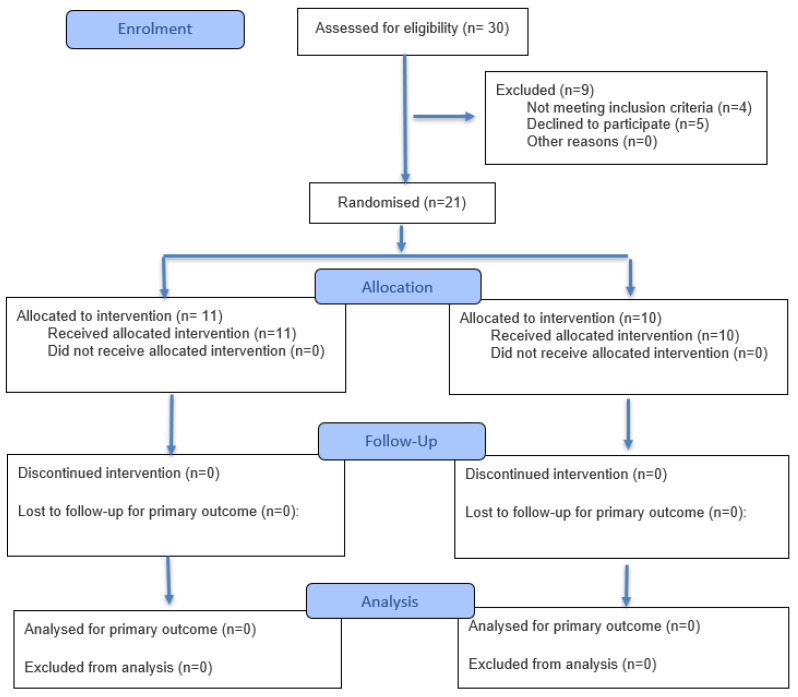
CONSORT 2025 Flow Diagram.

**Figure 3 jfmk-10-00433-f003:**
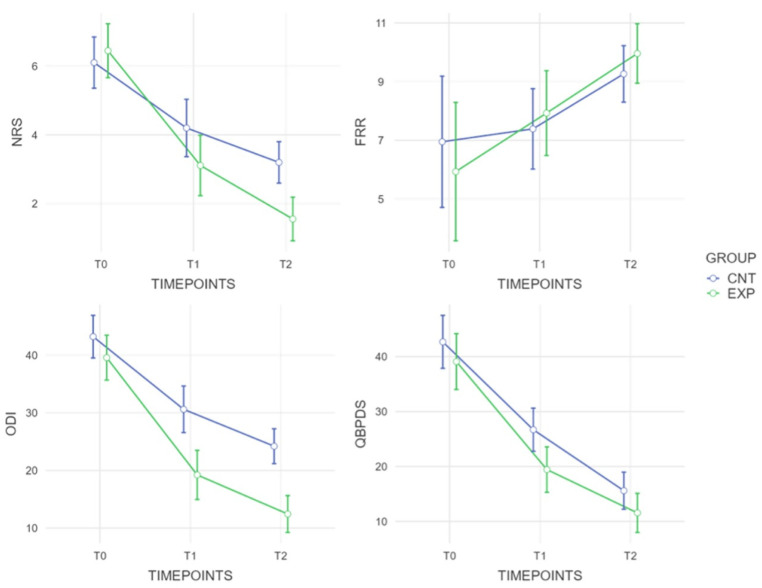
Graphic description of the outcome measures assessed in the different time points.

**Table 1 jfmk-10-00433-t001:** Outcome measures assessed in the different time points.

Outcome	Group	T0			T1			T2			Group × Time Interaction *p*-Value
FRR	EXP	5.93	±	3.174	7.93	±	2.246	9.96	±	1.316	
	CNT	6.95	±	3.506	7.39	±	1.865	9.26	±	1.545	0.942 (η^2^_gen_ = 0.039)
	between group	*p* = 0.51, MD = 1.018, ES: 0.303			*p* = 0.57, MD = −0.53, ES: −0.261			*p* = 0.31, MD = −0.69, ES: −0.480			
ODI	EXP	39.56	±	7.178	19.22	±	6.22	12.44	±	2.555	
	CNT	43.2	±	3.458	30.6	±	5.892	24.2	±	5.75	<0.001 (η^2^_gen_ =0.107)
	between group	*p* = 0.17, MD = 3.644, ES: 0.659			*p* < 0.001, MD = 11.37, ES: 0.883			*p* < 0.001, MD = 11.75, ES: 0.591			
QBPDS	EXP	39.11	±	6.864	19.44	±	2.186	11.56	±	5.79	
	CNT	42.7	±	7.543	26.7	±	7.818	15.6	±	4.3	0.035 (η^2^_gen_ = 0.049)
	between group	*p* = 0.29, MD = 3.589, ES: 0.496			*p* = 0.02, MD = 7.25, ES: 0.731			*p* = 0.1, MD = 4.04, ES: 0.820			
NRS	EXP	6.44	±	0.882	3.11	±	1.054	1.65	±	1.03	
	CNT	6.1	±	1.287	4.2	±	1.398	2.9	±	0.738	0.046 (η^2^_gen_ = 0.128)
	between group	*p* = 0.51, MD = −0.344, ES: −0.309			*p* = 0.07, MD = 1.08, ES: 0.575			*p* = 0.03, MD = 1.16, ES: 0.411			

Abbreviations: η^2^_gen_ = Generalized Eta Squared; FRR: Flexion Relaxation Ratio; ODI: Oswestry Disability Index; QBPDS: Quebec Back Pain Disability Scale; MD: mean difference; NRS: Numeric Rating Scale.

## Data Availability

The original contributions presented in this study are included in the article. Further inquiries can be directed to the corresponding author.

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
