# Peer review of "Effects of Pilates Matwork Core Exercises on Functioning in Middle-Aged Adult Women with Chronic Nonspecific Low Back Pain Through Flexion Relaxation Phenomenon Analysis: A Pilot RCT"

_jfmk, 2025, doi:10.3390/jfmk10040433_

Round 1
Reviewer 1 Report
Comments and Suggestions for Authors
The manuscript entitled “Effects of Pilates matwork core-exercises on functioning in middle adult women with chronic nonspecific low back pain through flexion relaxation phenomenon analysis” presents an interesting pilot study on the role of Pilates in chronic nonspecific low back pain (CNLBP). The paper is well-written, the methodology is adequately described, and the results are clearly presented. The topic is relevant, and the findings may contribute to the rehabilitation field.
Overall, the manuscript is of good quality and merits publication after minor revisions as outlined below:
Major Strengths
-
Clear description of intervention and control group protocols.
-
Appropriate use of validated outcome measures (ODI, QBPDS, NRS, FRR).
-
Novelty in assessing Pilates effects through FRP analysis.
-
Discussion well-linked with current literature and clinical implications.
Points Requiring Minor Revision
-
Abstract: Please specify the exact sample size in the methods section of the abstract (n = 21) for clarity.
-
Introduction: Some references are quite old (e.g., Floyd & Silver, 1955). Consider balancing with more recent literature on Pilates and FRP.
-
Methods:
-
Clarify the randomization process (how was allocation performed?).
-
The description of exercises is very detailed; this section could be slightly shortened or moved to a supplementary file for readability.
-
-
Results: In Table 1, ensure that effect sizes (ES) are consistently reported with the same decimal places.
-
Discussion: Expand slightly on the clinical implications of FRR exceeding the 9.5 cut-off only in the Pilates group, as this is a key finding.
-
Limitations: The limitations are appropriately mentioned, but please also acknowledge the lack of blinding as a potential source of bias.
-
Language/Style: Minor English editing is recommended to improve fluency (e.g., “non-inferior” could be replaced with “comparable” for readability).
Recommendation
I recommend Minor Revision. The manuscript will be suitable for publication once the above points are addressed.
Author Response
The manuscript entitled “Effects of Pilates matwork core-exercises on functioning in middle adult women with chronic nonspecific low back pain through flexion relaxation phenomenon analysis” presents an interesting pilot study on the role of Pilates in chronic nonspecific low back pain (CNLBP). The paper is well-written, the methodology is adequately described, and the results are clearly presented. The topic is relevant, and the findings may contribute to the rehabilitation field. Overall, the manuscript is of good quality and merits publication after minor revisions as outlined below:
Major Strengths
Clear description of intervention and control group protocols.
Appropriate use of validated outcome measures (ODI, QBPDS, NRS, FRR).
Novelty in assessing Pilates effects through FRP analysis.
Discussion well-linked with current literature and clinical implications.
We wish to express our sincere gratitude for the time and meticulous attention dedicated to the review of our manuscript. We are particularly appreciative of the precise and detailed assessment which so carefully highlighted the strengths of the contribution. We hereby confirm our commitment to resubmit the revised manuscript, accompanied by the following point-by-point response:
Points Requiring Minor Revision
- Abstract: Please specify the exact sample size in the methods section of the abstract (n = 21) for clarity.
Thank you very much for your comment, we have integrated it with the number of study participants as reported
- Introduction: Some references are quite old (e.g., Floyd & Silver, 1955). Consider balancing with more recent literature on Pilates and FRP.
Thanks for your comment, we've replaced the outdated references with a more recent one.
- Methods:
- Clarify the randomization process (how was allocation performed?).
Thanks for your comment, we added a paragraph on randomization process.
- The description of exercises is very detailed; this section could be slightly shortened or moved to a supplementary file for readability.
Thank you very much for your comment, the description of the exercises is actually a bit long-winded but it is necessary to describe the methods in detail.
- Results: In Table 1, ensure that effect sizes (ES) are consistently reported with the same decimal places.
Thank you very much for your comment, we reported same decimal places.
- Discussion: Expand slightly on the clinical implications of FRR exceeding the 9.5 cut-off only in the Pilates group, as this is a key finding.
Thanks for the helpful comment, we have highlighted the concept expressed in the discussion
- Limitations: The limitations are appropriately mentioned, but please also acknowledge the lack of blinding as a potential source of bias.
Thanks for the suggestion, we have added this additional limitation;
- Language/Style: Minor English editing is recommended to improve fluency (e.g., “non-inferior” could be replaced with “comparable” for readability).
Thanks for this very helpful comment, we have edited the sentences as suggested.
Reviewer 2 Report
Comments and Suggestions for Authors
Major Concerns
Sample Size and Justification: The study includes a total of 21 participants (approximately 10-11 per group). This is a tiny sample for a parallel-group RCT, even for a pilot study. The authors must:
Explicitly state that this is a pilot study in the title, abstract, and methods.
Provide a justification for the sample size (e.g., based on feasibility/practical constraints for a pilot study).
Discuss the limitations of the small sample size more thoroughly in the discussion, explicitly noting that it limits the statistical power and generalizability of the findings. A post-hoc power analysis might be considered.
Clarity and Consistency in Statistical Reporting:
The statistical analysis section is unclear. It mentions using independent t-tests, Mann-Whitney tests, and repeated measures ANOVA, but it's not specified which test was used for which outcome or for which part of the analysis (between-group vs. within-group).
Table 1 reports p-values for between-group comparisons, but it's ambiguous if these are from independent t-tests/Mann-Whitney at each time point or derived from a mixed-model ANOVA. The methods should clearly state the model used (e.g., a two-way mixed ANOVA with time as the within-subject factor and group as the between-subject factor).
The p-values in Table 1 are reported with inconsistent precision (e.g., p=0.51, p<.001). Standardise this to three decimal places (e.g., p = 0.510, p < 0.001).
Interpretation of Non-Significant FRP Results: The authors state that "Pilates has proven to be non-inferior to conventional treatment" regarding the FRR. This is a strong claim that requires a pre-specified non-inferiority margin and a specific statistical test for non-inferiority, which was not performed. The data simply shows no statistically significant difference. The language should be toned down to reflect this (e.g., "No statistically significant between-group differences in FRR were found...").
Specific Comments/Questions/Suggestions
Title and Abstract:
Title: Consider adding "A Pilot Study" to the title to manage reader expectations.
Abstract - Methods: Specify the total number of participants (N=21) and the group allocation ratio.
Abstract - Results: The phrase "allocated with a ratio 1:1 in the Pilates group, and in parallel in the control group" is grammatically awkward. Rephrase to "21 participants were randomly allocated to either the Pilates group (n=10/11) or the control group (n=11/10)."
Abstract - Conclusions: The conclusion "Pilates proved to be an effective rehabilitation method" is too strong for a pilot study. Consider "This pilot study provides preliminary evidence that Pilates may be an effective rehabilitation method..."
Introduction:
The introduction is generally well-written and establishes a clear rationale. It could be slightly strengthened by more precisely stating the primary and secondary hypotheses of the current study in the final paragraph.
Methods:
2.1 Participants:
Provide a flow diagram (CONSORT) showing recruitment, randomisation, allocation, follow-up, and analysis. This is crucial for transparency, especially with a small sample.
Clarify the randomisation procedure. How was the random sequence generated? How was the allocation concealed?
2.2 Intervention:
The control group exercises are well-described, but it's unclear if they were also supervised by a physiotherapist to the same extent as the Pilates group. This is a key detail for ensuring that the only major difference between groups was the type of exercise, not the amount of attention or supervision.
2.4 Outcome Measures:
The introductory sentence "Performance and scores on scales administered at the beginning and end of treatment improved:" is out of place in the Methods section and should be deleted, as it reports a result.
For the FRP assessment, provide more detail on the EMG data processing. How was the RMS calculated (window size, overlap)? The formula FRR = FLEX / MVF is stated, but the text later says the data was normalized to the upright phase. This is confusing. The exact, final formula used for the reported FRR must be explicitly stated.
Statistical Analysis:
As noted in Major Concern #2, this section needs a complete overhaul. Specify:
The specific statistical test used to check normality (e.g., Shapiro-Wilk).
For baseline comparisons, which test (t-test/Mann-Whitney) was used for continuous data?
For the primary analysis, state the specific model (e.g., "A two-way mixed ANOVA was used to analyze the effects of Time (T0, T1, T2) and Group (EXP, CNT) on ODI, QBPDS, NRS, and FRR scores.").
Describe how post-hoc tests were handled if significant interactions were found.
Report the effect sizes (e.g., partial eta-squared for ANOVA) in the results table or text, not just for between-group comparisons.
Results:
Table 1:
The table layout is confusing. The "RM-ANOVA" column seems to report a p-value, but it's unclear what this tests (the Time*Group interaction effect?). The column must be clearly labeled (e.g., "p-value for Group x Time Interaction").
Abbreviations in the table (MD, ES) should be defined in the table legend.
Provide the within-group data (mean ± SD) for all time points in a clear table. This allows readers to see the improvement over time within each group.
Text and Figure 2:
The results text frequently refers to a "Figure 2," but this figure was not included in the provided manuscript. Ensure all figures are present and that the text accurately describes them.
When stating "we showed a significant difference for perceived pain at T2," confirm that this p-value (p=0.03) is from a between-group comparison at T2 and that it survives correction for multiple comparisons if applicable.
Discussion:
Limitations: The limitations section is good but can be expanded.
Explicitly state the small sample size as the primary limitation.
Acknowledge the potential for Type II error (false negative) especially concerning the FRR results, given the small sample.
The lack of a true "no-treatment" or "sham" control group is a limitation, as both interventions may have non-specific effects.
The short-term follow-up (8 weeks) should be emphasized as insufficient to assess long-term efficacy.
Interpretation: Re-frame the claim of "non-inferiority" as discussed in Major Concern #3. Focus on the promising finding that only the Pilates group crossed the clinically important FRR threshold of 9.5 at follow-up, suggesting a potential unique neuromuscular benefit that warrants investigation in a larger trial.
Minor Editorial Comments:
Line 45 (Abstract): "Pilates has proven to be non-inferior..." -> See major concern #3.
Line 144: "The eight selected exercises will be performed..." -> Should be in past tense: "were performed..."
Throughout: Ensure consistent use of "sEMG" or "surface electromyography."
References: Check reference formatting for consistency with the journal's guidelines. For example, reference 27 (Tottoli et al.) appears to be a very recent (2024) and highly relevant study, which is good to see.
Author Response
Major Concerns
Sample Size and Justification: The study includes a total of 21 participants (approximately 10-11 per group). This is a tiny sample for a parallel-group RCT, even for a pilot study. The authors must:
Explicitly state that this is a pilot study in the title, abstract, and methods.
Thanks for the meaningful comment, we have made the requested changes, as recommended.
Provide a justification for the sample size (e.g., based on feasibility/practical constraints for a pilot study).
Thank you for the insightful comment. Assuming an expected effect size (Cohen's d) of 0.80 based on previous literature, with alpha = 0.05 (two-tailed), the study required a target sample of 30 participants (15 per group). Due to unforeseen recruitment constraints, the final sample included 21 participants (11 in exp group and 10 in cnt group).
Discuss the limitations of the small sample size more thoroughly in the discussion, explicitly noting that it limits the statistical power and generalizability of the findings. A post-hoc power analysis might be considered.
Thanks for the comment, it helped us understand this further limitation of the study that we have made known.
Clarity and Consistency in Statistical Reporting:
The statistical analysis section is unclear. It mentions using independent t-tests, Mann-Whitney tests, and repeated measures ANOVA, but it's not specified which test was used for which outcome or for which part of the analysis (between-group vs. within-group). Table 1 reports p-values for between-group comparisons, but it's ambiguous if these are from independent t-tests/Mann-Whitney at each time point or derived from a mixed-model ANOVA. The methods should clearly state the model used (e.g., a two-way mixed ANOVA with time as the within-subject factor and group as the between-subject factor).
Thanks for the comment, we revised the text accordingly.
The p-values in Table 1 are reported with inconsistent precision (e.g., p=0.51, p<.001). Standardise this to three decimal places (e.g., p = 0.510, p < 0.001).
Thanks for the comment, we revised Table 1 accordingly.
Interpretation of Non-Significant FRP Results: The authors state that "Pilates has proven to be non-inferior to conventional treatment" regarding the FRR. This is a strong claim that requires a pre-specified non-inferiority margin and a specific statistical test for non-inferiority, which was not performed. The data simply shows no statistically significant difference. The language should be toned down to reflect this (e.g., "No statistically significant between-group differences in FRR were found...").
Thanks for highlighting this factor in your comment, we've added it to the results.
Specific Comments/Questions/Suggestions
Title and Abstract:
Title: Consider adding "A Pilot Study" to the title to manage reader expectations.
We've added your suggestion to the comment as previously noted, thank you.
Abstract - Methods: Specify the total number of participants (N=21) and the group allocation ratio.
Thanks for your comment, we've added the number of study participants to the abstract.
Abstract - Results: The phrase "allocated with a ratio 1:1 in the Pilates group, and in parallel in the control group" is grammatically awkward. Rephrase to "21 participants were randomly allocated to either the Pilates group (n=10/11) or the control group (n=11/10)."
Thanks for the comment, we've rephrased the sentence.
Abstract - Conclusions: The conclusion "Pilates proved to be an effective rehabilitation method" is too strong for a pilot study. Consider "This pilot study provides preliminary evidence that Pilates may be an effective rehabilitation method..."
Thanks for the comment, we fixed it
Introduction. The introduction is generally well-written and establishes a clear rationale. It could be slightly strengthened by more precisely stating the primary and secondary hypotheses of the current study in the final paragraph.
Thanks for the comment, actually the second objective that we added had not been made explicit
Methods:
2.1 Participants:
Provide a flow diagram (CONSORT) showing recruitment, randomisation, allocation, follow-up, and analysis. This is crucial for transparency, especially with a small sample. Clarify the randomisation procedure. How was the random sequence generated? How was the allocation concealed?
Thank you, the flow of participants throughout the trial, including enrollment, allocation, follow-up, and analysis, is detailed within the accompanying CONSORT flowchart, as depicted in Figure 1. Moreover, we added the procedure of randomization.
2.2 Intervention:
The control group exercises are well-described, but it's unclear if they were also supervised by a physiotherapist to the same extent as the Pilates group. This is a key detail for ensuring that the only major difference between groups was the type of exercise, not the amount of attention or supervision.
Thank you for pointing this out, we have and emphasized the therapist's supervision also in the control
2.4 Outcome Measures:
The introductory sentence "Performance and scores on scales administered at the beginning and end of treatment improved:" is out of place in the Methods section and should be deleted, as it reports a result.
Thanks for the suggestion, we have removed the sentence
For the FRP assessment, provide more detail on the EMG data processing. How was the RMS calculated (window size, overlap)? The formula FRR = FLEX / MVF is stated, but the text later says the data was normalized to the upright phase. This is confusing. The exact, final formula used for the reported FRR must be explicitly stated.
Thanks for the helpful comment, we've both added more description of the RMS calculation and removed misleading statements.
Statistical Analysis:As noted in Major Concern #2, this section needs a complete overhaul. Specify:The specific statistical test used to check normality (e.g., Shapiro-Wilk).
Thank you, we stated specific statistical test used to check normality
For baseline comparisons, which test (t-test/Mann-Whitney) was used for continuous data?
Indipendent t-test.
For the primary analysis, state the specific model (e.g., "A two-way mixed ANOVA was used to analyze the effects of Time (T0, T1, T2) and Group (EXP, CNT) on ODI, QBPDS, NRS, and FRR scores.").
Thank you for pointing out, we omitted, but we performed a two-way mixed repeated measure ANOVA.
Describe how post-hoc tests were handled if significant interactions were found.
Post-hoc comparisons were performed using Holm correction applied
Report the effect sizes (e.g., partial eta-squared for ANOVA) in the results table or text, not just for between-group comparisons.
Thanks for your insightful comment, we reported the η²gen
Results:
Table 1:
The table layout is confusing. The "RM-ANOVA" column seems to report a p-value, but it's unclear what this tests (the Time*Group interaction effect?). The column must be clearly labeled (e.g., "p-value for Group x Time Interaction").
Thanks for your insightful comment, we edited the subheading, as recommended
Abbreviations in the table (MD, ES) should be defined in the table legend.
We clarified the abbreviations.
Provide the within-group data (mean ± SD) for all time points in a clear table. This allows readers to see the improvement over time within each group.
Thank you for pointing out, we reported all data groups for each timepoints
Text and Figure 2:
The results text frequently refers to a "Figure 2," but this figure was not included in the provided manuscript. Ensure all figures are present and that the text accurately describes them.
Thank you, we have successfully inserted the necessary CONSORT Flowchart, which has been designated as Figure 2 to reflect its role in documenting participant enrollment. Consequently, the figure previously labeled as Figure 2 has been correctly renumbered as Figure 3 within the manuscript.
Discussion:
Limitations: The limitations section is good but can be expanded.
Thanks, we've expanded the limitations section as suggested.
Explicitly state the small sample size as the primary limitation.
Thanks for the comment, the low sample size was already made explicit as the first limitation, we have added the discussion on generalizability and statistical power.
Acknowledge the potential for Type II error (false negative) especially concerning the FRR results, given the small sample.
Thanks, we've expanded the limitations section as suggested.
The lack of a true "no-treatment" or "sham" control group is a limitation, as both interventions may have non-specific effects.
Thanks for the suggestion, we have added this limitation as the fourth point
The short-term follow-up (8 weeks) should be emphasized as insufficient to assess long-term efficacy.
Thanks for the comment, we've added this suggestion to the limitations as well.
Interpretation: Re-frame the claim of "non-inferiority" as discussed in Major Concern #3. Focus on the promising finding that only the Pilates group crossed the clinically important FRR threshold of 9.5 at follow-up, suggesting a potential unique neuromuscular benefit that warrants investigation in a larger trial.
Thanks for the comment, we further emphasized the concept in the conclusions
Minor Editorial Comments:
Line 45 (Abstract): "Pilates has proven to be non-inferior..." -> See major concern #3.
Thanks for the comment, it has already been fixed as previously suggested
Line 144: "The eight selected exercises will be performed..." -> Should be in past tense: "were performed..."
Thanks for the comment, we have corrected it.
Throughout: Ensure consistent use of "sEMG" or "surface electromyography."
Thanks for reporting it
References: Check reference formatting for consistency with the journal's guidelines. For example, reference 27 (Tottoli et al.) appears to be a very recent (2024) and highly relevant study, which is good to see.
Thanks for the comment, in this regard we have replaced a reference 17 with a more recent reference